# Noggin-Loaded PLA/PCL Patch Inhibits BMP-Initiated Reactive Astrogliosis

**DOI:** 10.3390/ijms252111626

**Published:** 2024-10-29

**Authors:** James Hawes, Ana Gonzalez-Manteiga, Kendall P. Murphy, Marina Sanchez-Petidier, Victoria Moreno-Manzano, Bedika Pathak, Kristin Lampe, Chia-Ying Lin, Jose L. Peiro, Marc Oria

**Affiliations:** 1Center for Fetal and Placental Research, Cincinnati Children’s Hospital Medical Center (CCHMC), Cincinnati, OH 45229, USA; hawesjs@mail.uc.edu (J.H.); kendallhuddleston2015@gmail.com (K.P.M.); bp918922@ohio.edu (B.P.); kristin.lampe@cchmc.org (K.L.); jose.peiro@cchmc.org (J.L.P.); 2Department of Radiation Oncology, University of Cincinnati College of Medicine, Cincinnati, OH 45219, USA; gonza2aa@ucmail.uc.edu; 3Neuronal and Tissue Regeneration Laboratory, Prince Felipe Research Institute, 46512 Valencia, Spain; marinas@externas.sescam.jccm.es (M.S.-P.); vmorenom@cipf.es (V.M.-M.); 4Neuronal Circuits and Behaviour Group, Hospital Nacional de Parapléjicos, Instituto de Investigación Sanitaria de Castilla-La Mancha (IDISCAM), 45071 Toledo, Spain; 5Convergent Bioscience and Technology Institute, Department of Biomedical Engineering and Informatics, Indiana University, Indianapolis, IN 46202, USA; cyjlin@iu.edu; 6Department of Surgery, University of Cincinnati College of Medicine, Cincinnati, OH 45219, USA; 7University of Cincinnati Cancer Center (UCCC), Cincinnati, OH 45219, USA; 8University of Cincinnati Brain Tumor Center (BTC), Cincinnati, OH 45219, USA

**Keywords:** noggin, PLA/PCL polymer smart patch, myelomeningocele, astrogliosis, neural precursor cells

## Abstract

Myelomeningocele (MMC) is a congenital birth defect of the spine and spinal cord, commonly treated clinically through prenatal or postnatal surgery by repairing the unclosed spinal canal. Having previously developed a PLA/PCL polymer smart patch for this condition, we aim to further expand the potential therapeutic options by providing additional cellular and biochemical support in addition to its mechanical properties. Bone morphogenetic proteins (BMPs) are a large class of secreted factors that serve as modulators of development in multiple organ systems, including the CNS. We hypothesize that our smart patch mitigates the astrogenesis induced, at least partly, by increased BMP activity during MMC. To test this hypothesis, neural stem or precursor cells were isolated from rat fetuses and cultured in the presence of Noggin, an endogenous antagonist of BMP action, with recombinant BMPs. We found that the developed PLA/PCL patch not only serves as a biocompatible material for developing neural stem cells but was also able to act as a carrier for BMP–Notch pathway inhibitor Noggin, effectively minimizing the effect of BMP2 or BMP4 on NPCs cultured with the Noggin-loaded patch.

## 1. Introduction

Spina bifida (SB) is the most common congenital central nervous system (CNS) disorder of the spinal cord, resulting from incomplete closure of the neural tube during embryogenesis. This condition encompasses both open and closed neural tube defects (NTDs), with myelomeningocele (MMC) being the most prevalent type. MMC is characterized by exposed neural tissue and brain/spinal cord malformations, where a portion of the spinal cord and surrounding tissues protrudes without any coverage through an opening in the vertebrae, resulting in a visible sac or lesion on the skin [1]. Since the neural tube forms early in fetal development and gives rise to the brain and spinal cord, MMC is a serious condition that can cause a range of physical and neurological impairments [2]. The severity of symptoms varies depending on the location and size of the lesion, but common complications include paralysis or weakness in the legs, loss of bladder and bowel control, hindbrain herniation, hydrocephalus, and intellectual disability [1,2,3].

Comprehensive evaluation and care, including modern surgical techniques, are crucial for improving outcomes in SB cases. Multidisciplinary approaches are essential to enhance mobility and prevent complications [2,3,4]. The current standard treatment for MMC after an accurate antenatal diagnosis is prenatal surgery at 23–25 weeks’ gestation to close the lesion and protect the spinal cord from further damage [2,4,5]. The goal of surgery is to prevent additional damage to the exposed spinal cord, reduce the risk of infection, avoid cerebrospinal fluid (CSF) leakage, and improve or reverse Chiari type II malformations. Various surgical techniques are available, including primary closure, which involves suturing the skin edges of the lesion together after release of the spinal cord and duraplasty, and different forms of tissue or grafting using patches [6,7,8,9,10]. Fetal surgery for prenatal MMC repair is associated with better postnatal outcomes, including a reduced need for shunting and improved motor function; however, it also carries increased risks for maternal–fetal morbidity, including prematurity, membrane rupture, uterine rupture, dehiscence, and maternal hemorrhage [11,12,13,14,15]. Some of these complications have been reduced by the implementation of a fetoscopic, minimally invasive approach in the last few years.

To expand the therapeutic potential of MMC repair, extensive studies have examined the effects of MMC lesions on the developing spinal cord, particularly focusing on resident neural progenitor cells (NPCs) [7,9,10]. Multipotent NPCs are crucial for neural development, giving rise to neurons, oligodendrocytes, and astrocytes. As such, NPCs represent a promising pool for regenerative therapies [16]. Additionally, stem cell engrafting has emerged as a therapeutic strategy for spinal cord injury by a paracrine function releasing trophic and growth factors, creating a restorative environment for tissue regeneration [17,18,19].

Understanding how NPCs are affected in MMC is pivotal for developing novel and effective stem cell therapies for tissue regeneration in SB. Our recent research reports premature astrogliosis and altered NPC presence in SB models, leading to neuronal dysfunction [7]. Indeed, NPCs undergo early astrocytic reactions and exhibit altered expressions of *Pax6*, *Olig2*, and *Nkx2.2* in experimental induced SB models. This alteration correlates with elevated genes involved in the Notch–BMP pathway, a molecular mechanism involved in cell fate determination and differentiation [7,20,21,22]. Hence, manipulating the Notch–BMP pathway, in addition to creating a mechanical barrier, may improve neurological function following delivery [14,15].

Therefore, protecting natural neural tube development by manipulating the roles and ultimate lineage of NPCs requires strategies that address both the mechanical and biological needs of the injured tissue [23]. Mechanically, this includes creating a watertight barrier to address the lesion characteristic of MMC by a surgical intervention early in gestation. On a molecular and cellular level, novel treatments aim to reverse the progressive alterations in the spinal cord, addressing the earlier effects in early development. Our bioengineering research laboratory has developed a poly(L-lactic acid, PLA) and poly(ε-caprolactone, PCL) blended ‘smart’ patch capable of self-expansion for potential MMC repair, which can be used independently or in combination with cell/drug therapy to address both the physical closure and restorative processes in the damaged tissue [24,25]. As previously shown, these PCL/PLA patches have a glass transition temperature of 37.6 °C to facilitate their use in laparoscopic repair. The patches demonstrate a permeability to water vapor and liquid water of 0.000206 and 0.000414 µL/cm^2^/min, respectively, while in-vitro degradation studies of these patches in both PBS and human amniotic fluid displayed progressive hydrolytic degradation and a 16-week weight loss of 13.5% and 14.9%, respectively. There are no significant weight or dimensional changes of the patch upon submersion, confirming no evidence of swelling and that the patch self-expansion is due primarily to shape recovery [24]. Additionally, in vivo studies of the patch in a rat model suggested no adverse or inflammatory reactions to the patch in either subcutaneous implantation or as a dural substitute. When implanted in the neural tissue, there were no signs of astrogliosis nor were there functional alterations of neuron potential amplitude after 4 weeks of observation [25]. In this report, we combined NPCs with PLA/PCL patches to offer a novel therapeutic strategy for MMC, addressing both mechanical closure and regenerative processes. After validating the biocompatibility of the PLA/PCL patches for NPCs, we evaluated whether loading the patches with Noggin, a classical inhibitor of the Notch–BMP signaling pathway, could modulate NPC differentiation towards an astrocytic lineage. Our results demonstrated that Noggin-loaded PLA/PCL patches could restrict astrocytic differentiation of NPCs by inhibiting BMP signaling and maintain these NPCs with the potential to create neurons.

## 2. Results

### 2.1. Biocompatibility Validation of PLA/PCL Patches in NPC Cultures

The main goal of this work is to further characterize the combination of NPC cultures and PLA/PCL patches previously developed by our group [24,25] as a potential therapeutic strategy in the context of MMC. In this regard, we first tested whether PLA/PCL patches could be biocompatible for NPC cultures.

To achieve this goal, we cultured NPCs from fetal rat spinal cord (E15) on PLA/PCL patches for 3 days in vitro. Using classical stem cell markers, we observed that NPCs remained viable on the patches (Figure 1A). Moreover, quantification of SOX2, Ki67, and Nestin-positive cells indicated that NPCs remained undifferentiated on the patches, suggesting these cells may maintain pluripotency onto the patch (Figure 1B). Thus, these data suggest that the patch does not promote the early differentiation of NPCs, which may contribute to its potential therapeutic application in MMC.

Next, we evaluated whether an additional coating material could enhance NPC adherence to the patch. To this end, we seeded NPCs on patches with a classical coating material (Matrigel diluted 1:10) and without a coating (referred to as control) for 24 h, enough time to attach to a coated surface. The cells were fixed and immunostained for TUJ1 and GFAP; it was shown that the NPCs were able to grow and adhere to the coated patches, as expected (Figure 2). We observed no differences in the percentage of TUJ1+ cells between the coated and control patches. However, there was a significant increase in GFAP+ cells when NPCs were cultured on Matrigel-coated patches (Figure 2A,C). The data highlight the biocompatibility of the PLA/PCL patches for NPCs, suggesting their potential use in cell therapy for MMC.

To further validate the biocompatibility of the PLA/PCL patch, primary rat neuronal cultures (E14.5) were cultured on the poly-L-lysine-coated patch. Cortical neuronal cultures showed adequate TUJ+ cell density and morphology after 3 days in vitro, suggesting that neuronal cells can also be cultured on the PLA/PCL patches (Figure 2B,C).

### 2.2. Characterization of BMP2 and BMP4 Effect in 3D Neuroprogenitor Cell Cultures

Recent studies described early astrogenesis in animal models of SB [20]. Furthermore, it has been also reported that the activation of BMP signaling promotes an early differentiation of NPCs towards the astrogenic lineage [7,21,22]. Therefore, we next validated the effect of BMP2 and BMP4 exposure in our NPCs neurosphere cultures. 

As expected, NPCs displayed a stem cell-like behavior, forming spherical like-structures in less than 24 h in vitro and nearly all the cells formed increasingly larger spheres within the first 48 h in vitro, defined by a round shape, a dense core, and a clear periphery. Neurospheres with a spherical radius exceeding 100 µm exhibited a darker central region after 5–7 days in vitro, indicative of an increased cell density attributable to impaired growth factor perfusion within the inner regions (Figure 3A).

Once the neurospheres were formed, they were treated with BMP2 (50 ng/mL) or BMP4 (50 ng/mL) for 3 days for immunostaining and gene expression analysis. It is noteworthy that irregularly shaped clusters were formed simultaneously, mainly in cultures exposed to BMP2 and BMP4 (Figure 3B, right column, arrowheads). Indeed, this phenomenon may be correlated with a potential cell differentiation towards a glial cell lineage, as previous studies have shown [26].

To better ascertain whether BMP signaling may promote neural precursors differentiation, 3D neurosphere cultures were immunostained with Nestin and GFAP antibodies. In this regard, both BMP2 and BMP4 did not affect cell viability compared to the control but did promote significant increases in GFAP (Figure 4A). Particularly, GFAP intensity was mainly observed along the neurosphere borders exposed to BMP2, whereas for BMP4-treated neurospheres, the GFAP intensity increased, but diffusely inside the neurosphere (Figure 4A, middle and bottom rows).

Given that GFAP serves as both a classical indicator of astrogenic response and an early marker for neural stem cells, we assessed the gene expression of *Olig2* and *Nestin* to probe the potential influence of BMP exposure on alternative differentiation pathways. Notably, NPCs subjected to BMP2 and BMP4 culture conditions showed heightened levels of *Olig2* expression, albeit significantly lower than the *GFAP* expression. Consequently, these findings suggest a propensity of BMP to facilitate NPC differentiation. Notably, this differentiation primarily favors an astrogenic lineage rather than a gliogenic/oligodendrogenic lineage (Figure 4B).

### 2.3. Noggin-Loaded PLA/PCL Patches Restrict Astrogliosis in NPCs Cultures

An exacerbated astrogenic response is a classic feature of several types of spinal cord injury, including SB, and correlates with limited neural regeneration [27,28]. Considering that BMP2 and BMP4 treatment enhances the differentiation of neural progenitors towards the astrogenic lineage [20,29], inhibition of the BMP signaling pathway could be a potential therapeutic target in the context of MMC. In this regard, we used Noggin as a BMP signaling inhibitor to further investigate whether Noggin-loaded PLA/PCL patches can restrict NPC differentiation.

To first determine whether the patch could be loaded with Noggin, we preloaded a patch with a high concentration of Noggin (5 µg/mL) for 30 min, followed by a 48 h drying period. We then investigated the release profile of Noggin for the Noggin-loaded (NL) patch in 2 mL of growth media over 24 h (Figure 5A). The majority of Noggin was released within 2 h, reaching an average concentration of 6.63 ± 0.58 ng/mL. The rate of release continued to decay up to the 24 h timepoint, reaching a final concentration of 8.351 ± 0.68 ng/mL.

As shown in Figure 3A, 3D NPC neurospheres were then cultured with the patches for 3 days in vitro under various experimental conditions, as follows: control (non-loaded patch + NPCs); BMP4 (non-loaded patch + NPCs + BMP4); BMP4 + Noggin patch (non-loaded patch + NPCs + BMP4 + Noggin in solution); and BMP4 + NL patch (Noggin-loaded patch + NPCs + BMP4). Additionally, we included two distinct experimental groups to determine whether the PLA/PCL smart patch could act as a carrier for Noggin. In the BMP4 + Noggin group, NPCs were cultured with non-loaded patches with Noggin (100 ng/mL) and BMP4 directly added to the cell suspension. In the BMP4 + NL group, the patches were preloaded with Noggin as described above, but BMP4 was still added directly to the cell suspension. In all relevant groups, the growth factors and/or patch were added at the same timepoint. 

After 3 days in vitro, NPCs were collected from the patches for immunocytochemistry and qRT-PCR to analyze GFAP, Nestin, and Pax6 expression. As shown in Figure 5B, NPCs cultured on Noggin-loaded patches exhibited similar morphology and staining for Nestin, GFAP, and Pax6, regardless of BMP treatment. 

To further evaluate the effect of Noggin treatment on NPC-cultured patches, we also analyzed the gene expression of *GFAP*, *Nestin*, and *Pax6.* NPCs on PLA/PCL patches treated with BMP4 showed an increase in *GFAP* and *Pax6* gene expression compared to the control group (Figure 5C). These data suggest that BMP signaling drives NPC differentiation towards the astrogenic lineage on the patches, as previously demonstrated in the 3D culture model (Figure 4B). However, Noggin-treated patches restored both *GFAP* and *Pax6* gene expression to basal levels, indicating that inhibiting BMP signaling via Noggin-loaded patches might return NPC development to a physiological state (Figure 5C). Thus, these results suggest that BMP inhibition by Noggin may help to maintain the stemness properties of NPCs, potentially contributing to cell regenerative therapy for MMC.

Interestingly, we observed that Noggin-loaded patches effectively delivered Noggin to NPCs, thereby limiting the effect of BMP4 by decreasing *GFAP* and *PAX6* expression, as seen in the patches treated with Noggin directly in the culture media. Therefore, this result illustrates the potential of PLA/PCL smart patches as therapeutic carriers for drug-delivery strategies.

Overall, our data highlight the relevance of a novel approach using PLA/PCL patches not only as a mechanical barrier for the treatment of SB, but also as a potent drug carrier. Our results demonstrate that Noggin-loaded patches are capable of drug delivery and thereby partially restrict NPC differentiation toward the astrogenic lineage, thereby restoring physiological NPC development. Thus, the combination of bioengineering and stem cell therapy may provide a promising paradigm for the treatment of MMC. Consequently, the potential of this tool needs to be thoroughly investigated in the biology of NPCs and in the context of neural regeneration and tissue repair.

## 3. Discussion

MMC is the most challenging and severe congenital malformation of the CNS for which there is still no cure [27]. Although the etiology of MMC remains poorly understood, numerous efforts have been made to develop therapeutic solutions, primarily focusing on surgical procedures to close the spinal canal [29,30]. However, these strategies are still insufficient to significantly improve the quality of life of patients, who continue to experience various disabilities throughout their lives, including bladder dysfunction, orthopedic impairments, and cognitive difficulties [31]. Therefore, it is crucial to develop novel therapeutic approaches that not only focus on mechanical closure of the canal but also enhance regenerative approaches to tackle the underlying biological processes in the injured area.

We have previously demonstrated the efficacy of our ‘smart patch’ to accommodate the needs of fetoscopy-based SB repair, including biodegradability, water tightness, and thermoresponsive shape memory [25], and shown that the patch polymer does not promote a broader immune response or activation of resident microglia within the neural tube but instead suggests permissiveness to neural growth. In this report, we further explored the biological potential of PLA/PCL polymer smart patches as a promising biomaterial for MMC treatment, functioning both as carriers for stem cell therapy and drug-delivery systems. Specifically, we evaluated the in vitro capability of Noggin-loaded PLA/PCL patches to inhibit BMP-mediated astrogliosis, a common characteristic of MMC. 

BMPs are members of the TGFβ superfamily of growth factors that play a critical role in cell differentiation and fate commitment during development [32,33,34]. Numerous studies have reported the effects of these molecules on various types of pluripotent cells from different tissues, including neural progenitors. In most cases, BMP2 and BMP4 have been shown to promote the differentiation of these progenitors into astrocytes in various animal species, which is consistent with the in vitro results presented in this study [35,36,37]. Consequently, targeting and modulating BMP signaling offers a promising therapeutic strategy for neural disorders such as MMC. 

Several studies have used different types of BMPs to test their roles in developmental processes. In terms of neural and non-neural fate determination, Zhu and colleagues demonstrated that BMP4 activation—via the mediator Med23—promotes astrocyte formation and inhibits neuronal differentiation through in vitro and in vivo approaches using mouse and zebrafish embryos [38]. Conversely, other studies have reported that BMP4 signaling stimulates neuronal differentiation in cortical progenitor cultures derived from embryonic mice [39,40]. Therefore, future research is needed to elucidate the molecular pathways that control cell fate decisions in neural progenitors.

Nevertheless, discussing the most effective molecule to drive astrogliogenesis is beyond the scope of this work, since the main goal of this study was to evaluate the therapeutic properties of our PLA/PCL smart patch in vitro, rather than to investigate the mechanisms underlying astrogliogenesis through BMP signaling and in which BMP exerts a stronger effect. Here, we showed that Noggin-loaded patches reduced GFAP expression in BMP4-treated NPCs, demonstrating that our loaded patch could be an interesting tool to modulate the astrogliogenic effect driven by BMP signaling. 

In this regard, by using NPCs as an in vitro cellular model, we demonstrated, as follows, that: (1) the patches are biocompatible for NPCs; and (2) Noggin-loaded PLA/PCL patches restricted NPC differentiation towards an astrogenic lineage by inhibiting BMP signaling. 

First, we evaluated the biocompatibility of the patches with NPCs and demonstrated that they support NPC culture without altering cell morphology or fate. The results highlight the significant advantages of this biomaterial for cell therapy, allowing for specific cell lines such as NPCs to be cultured directly in the injured region, advancing a challenging but promising clinical approach based on drug-delivered cell transplantation in MMC.

We tested this concept using a patch loaded with Noggin to interfere with the Notch/BMP pathway of NPCs in vitro, a well-known interaction that significantly reduces astrogliosis and GFAP expression. Theoretically, this should also minimize the effects of amniotic fluid exposure on spinal cord NPC cell fate in vivo, ultimately decreasing the number of NPCs that commit to an astrocyte lineage. The PLA/PCL patch displayed no signs of cytotoxicity or inflammatory effects in vitro and in vivo in previous studies. Furthermore, the solvent used in preparing the patch is fully evaporated prior to any modifications and use; therefore, we have no reason to suspect that the patch adversely affects Noggin during the loading phase [25]. While introducing Noggin to minimize the astrogenic effects of MMC is not innovative, this study demonstrates the potential of our patch to act as a delivery device and provide both mechanical and biochemical therapies simultaneously. Our results suggest that this new patch will help advance fetoscopic approaches to prenatal management of the SB defect by not only preserving but also improving natural neural tube development, ultimately promoting the long-term motor function of patients. This is in stark contrast to current treatments, which primarily act as mechanical therapy, or in some cases promote tissue growth to naturally seal the defect, as opposed to biochemical treatments that preserve and regenerate the integrity of the spinal cord.

Unfortunately, this study is limited by the inability to adequately control long-term drug delivery, an essential requirement for more complex therapies addressing neural tube development. Specifically, the needs of the developing spinal cord are known to change at different stages of development, and the PCL/PLA patch, as presented, cannot currently accommodate those changes. Future work is ongoing to address this issue by providing a more controlled and potentially delayed delivery system within the patch, facilitating a more complex drug therapy regimen.

In conclusion, this smart PLA/PCL patch presents a promising paradigm by combining, as follows: (1) mechanical properties to physically close the spinal canal; (2) cell engraftment therapy to enhance regenerative processes; and (3) in situ drug delivery to specifically target signaling pathways involved in tissue repair. The combination of biomaterials and stem cell therapy proposed in this work suggests a promising avenue for novel regenerative strategies for CNS injuries [29,30]. Our PLA/PCL patch not only may provide mechanical support to physically close the spinal canal but also may serve as an effective carrier for engrafting NPCs and contributing to a regenerative environment in the injured area [31] Therefore, future experiments are needed to further characterize the potential and capability of this biomaterial, aiming to develop novel and efficient therapeutic strategies for SB.

## 4. Materials and Methods

The experimental protocols agreed with the National Institutes of Health Guidelines for Care and Use of Laboratory Animals and were approved by the Institutional Animal Care and Use Committee at Cincinnati Children’s Hospital Medical Center (IACUC 2019-0081).

### 4.1. NPC Isolation and Culture Covering Materials

Neural precursor cells (NPC) were harvested from E15 rat spinal cords, dissected in cold Hank’s balanced saline solution (HBSS) supplemented with penicillin–streptomycin. Dissected tissue was mechanically dissociated by gentle pipetting. NPCs were isolated and cultured as neurospheres in ultra-low attachment plates with the growing medium (growing conditions) NeuroCultTM Proliferation Medium (Stemcell Technologies, Grenoble, France), supplemented with, as follows: NeuroCultTM Proliferation Supplement (Stemcell Technologies); 100 U/mL penicillin (DE17-602E, Sigma, ST. Quentin Fallavier Cedex, France); 100 µg/mL streptomycin (Sigma); 0.7 U/mL heparin (H3393, Sigma); 20 ng/mL epidermal growth factor (EGF; 10605-HNAE, Thermo Fisher, Horsham, UK); and 20 ng/mL basic fibroblast growth factor (bFGF; 10014-HNAE, Invitrogen, Waltham, MA, USA). Cells were stored at 1 × 10^6^ cells/mL in CryoStor cell cryopreservation media (C2874, Millipore Sigma) at −80 °C.

### 4.2. NPC Culture and Treatments

NPC-forming neurospheres were disaggregated with Accutase (StemPro™ Accutase™, Cat. A1110501, Thermo Fisher) following the manufacturer’s instructions and seeded at a density of 2 × 10^5^ cells. Cells were seeded from storage and allowed to grow into spheres in the previously described growing medium for 7 days prior to any cytokine treatment. Brightfield images were taken at days 2 and 5 in vitro to ensure the formation and morphology of the neurospheres. 

To test the biocompatibility of the PLA/PCL patch in NPC cultures, 4 × 4 mm^2^ PLA-PCL patches were precoated with Matrigel (1:10). Cells were suspended in 5 mL of the medium and deposited on top of the patch surrounded by the medium; however, floating or immersion was avoided to encourage cell attachment during 30 min incubation at 37 °C. Then, fresh medium was added and cells were incubated for 3 days. 

To evaluate the biocompatibility on another cell type, rat cortical neurons were isolated from the cerebral cortex of E-15 Sprague–Dawley rats. Experimental procedures involving animal experimentation were previously approved by the Animal Care and Use Committee of the Centro de Investigación Príncipe Felipe (approved procedure: 2018/VSC/PEA/0058). Briefly, the brain cortex was mechanically disaggregated to facilitate the following enzymatic disaggregation with trypsin–EDTA (0.05%, 25300054, Gibco, Grand Island, NY, USA) at 37 °C for 10 min. DMEM high glucose (Cytiva HyClone™, Thermo Fisher) supplemented with 10% Fetal Bovine Serum (FBS) and 100 µg/mL penicillin/streptomycin (P/S) were used to dilute the trypsin. Then, they were filtered with a sterile cell strainer (70 µm, Corning Falcon™, New York, NY, USA) and plated directly on poly-L-lysine (PLL)-coated bottom-glass 24-well plates (Cellvis; Mountain View, CA, USA; #P24-0-N). After 2 h of incubation at 37 °C, 5% CO_2_, the plating medium was replaced by neurobasal medium supplemented with 1X B27 supplement (Gibco), 50 mM Glutamax (Gibco) and 100 µg/mL P/S.

To evaluate the BMPs and Noggin effect on 3D neurosphere culture, NPCs were cultured after 7 days in vitro with the described growing media additionally supplemented with 2% FBS, Noggin (100 ng/mL), BMP2 (50 ng/mL) or BMP4 (50 ng/mL), as appropriate, for 3 days prior to immunocytochemical or RT-qPCR analysis. Analysis was performed after 10 total days in vitro. 

### 4.3. PLA/PCL Patch Fabrication and Noggin Loading

Thin films of poly L-lactic acid (PLA) and polycaprolactone (PCL) (referred to as PLA/PCL patch for now on) were prepared by solvent casting, as previously described [24,25]. Briefly, 1.67 mg of PLA pellets (4032D, NatureWorks LLC, Plymouth, MN, USA) and 0.33 mg of PCL microspheres (CapaTM6506, Perstorp UK Ltd., Malmö, Sweden) were procured and dissolved in 20 mL of chloroform solvent (ACS grade, Labchem, Pittsburgh, PA, USA) in a closed tube. Once the compound was dissolved, the mix was cast in a mold until the chloroform totally evaporated, leaving the patch. 

To inhibit the BMP pathway, Noggin loading was performed by submerging 2 cm × 2 cm patches in highly concentrated Noggin (50 µg/mL) for 30 min. The patches were then allowed to dry in an incubator for 48 h before use in the experiments. Prior to the experiments, a Noggin release profile was determined using an ELISA (50-228-6693, Fishersci, Pittsburgh, PA, USA). This analysis was performed on 2 cm² Noggin-loaded patches that were submerged in a high concentration of Noggin (5 µg/mL) for 30 min and allowed to dry for 48 h. The Noggin-loaded (NL) patches were then placed in growth media for 24 h, and monitored at 1, 2, 4, 8, and 24 h time-points.

### 4.4. Immunocytochemistry

Sections approximately 5 µm in thickness were deparaffinized, rehydrated, and incubated in sodium citrate buffer (pH 6) for 30 min at 95 °C to retrieve antigens. Sections were permeabilized with 0.5% Triton X-100 (Sigma-Aldrich, St. Louis, MO, USA) in phosphate-buffered saline (PBS) and incubated in 3% peroxide for 15 min at room temperature. Non-specific binding was blocked for 1 h with 5% BSA in PBS at room temperature, and sections were then incubated overnight at 4 °C in a humidity chamber with the following primary antibodies: anti-GFAP (#AB4674, Chicken, Abcam, Cambridge, MA, USA) (1:500); anti-Pax6 (Abcam #ab5790, Rabbit) (1:1000); anti-Nestin (BD Biosciences, #556309, Rat, Franklin Lakes, NJ, USA) (1:50); anti-ki67 (ab15580, Abcam, Chicken) (1:600); anti-SOX2 (ab75179, Abcam, Rabbit) (1:300); and anti-β-III tubulin (MO15013, Neuromics, Edina, MN, USA, Mouse) (1:400). Sections were washed and incubated for 1 h with Alexa Fluor 488 (#1531671, Donkey and #1531669 Goat, #1990462, Goat), Alexa Fluor 568 (#1691230, Goat, #1398018, Goat, #1504529, Goat), or Alexa Fluor 647 (#1445259, Goat, #1608641, Donkey) conjugated secondary antibodies (Life Technologies, Carlsbad, CA, USA) (1:1000) in a dark humidity chamber at room temperature. Slides were washed, covered with mounting media containing DAPI (Southern Biotech, Birmingham, AL, USA), and visualized with a Nikon fluorescent microscope (Nikon Inc., Melville, NY, USA). Cell counts, DAPI, GFAP+, and Nestin+ analyses were conducted using NIS Elements AR 4.5 software (Nikon Instruments Inc., Melville, NY, USA). Quantification was conducted using more than eight random images from each culture group.

### 4.5. RT-qPCR

RNA was extracted using trizol and glycogen before being precipitated in isopropanol overnight. RNA quantity was assessed through spectrophotometric analysis using an Epoch Biotek Spectrophotometer (Biotek Instruments, Winooski, VT, USA). Utilizing the RT2 First Strand Kit (Qiagen Sciences, Germantown, MD, USA), 1 μg RNA/sample was reverse-transcribed into cDNA. A 1-μg cDNA sample was then used as a template for RT-qPCR employing TaqMan^®^ gene expression assays (Applied Biosystems, Foster City, CA, USA) in the 7500 Fast Real-Time PCR System. Samples were run in duplicate for target genes and were normalized using HPRT1 as an endogenous control. Relative quantification of transcript expression was performed using the 2^−ΔΔCt^ method, where Ct represents the threshold cycle.

### 4.6. Statistical Analysis

All statistical analyses and graphs were performed in Graph Pad Prism 9 software (GraphPad Software Inc., La Jolla, CA, USA). Differences among multiple groups were analyzed by one-way analysis of variance (ANOVA) using Tukey’s post hoc test. Results are reported as means ± standard error (SE) for the relative gene expression (2^−ΔΔCt^) and means ± standard deviation (SD) for all cell-counting analysis. A *p*-value < 0.05 was considered statistically significant.

## 5. Conclusions

We tested the therapeutic capability of a novel biomaterial previously developed in our group, PLA/PCL patches. In this work, we demonstrated that loading the PLA/PCL patches with Noggin, a classical inhibitor of the BMP pathway, reduced NPC differentiation towards an astrogenic lineage. Since exacerbation of the early astrogenic response is associated with SB and thus tissue damage, understanding the potential of this tool could be a potentially translatable approach for MMC treatment.

## Figures and Tables

**Figure 1 ijms-25-11626-f001:**
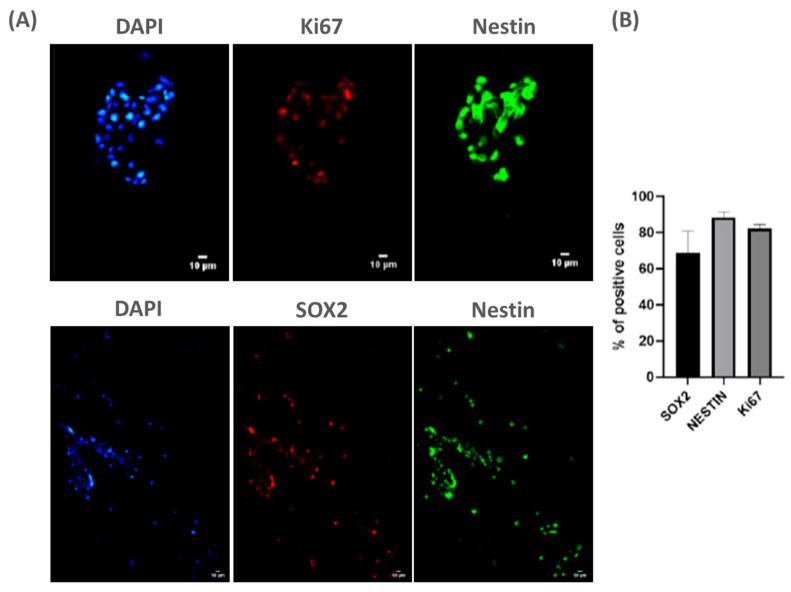
NPCs are viable and do not differentiate when seeded over PLA/PCL: (**A**) representative panel showing NPCs stained with classic markers of stem cell progenitors (ki67, SOX2, and Nestin) on day 3 cultured on PLA/PCL patches. Scale bar: 10 µm; and (**B**) the graph shows the quantification of SOX2+, ki67+ and Nestin+ cells cultured on the PLA/PCL patch for 3 days. Of note, the vast majority of adherent cells continue to express the common stem cell markers SOX2, Nestin, and Ki67, indicating maintenance of pluripotency. n = 3.

**Figure 2 ijms-25-11626-f002:**
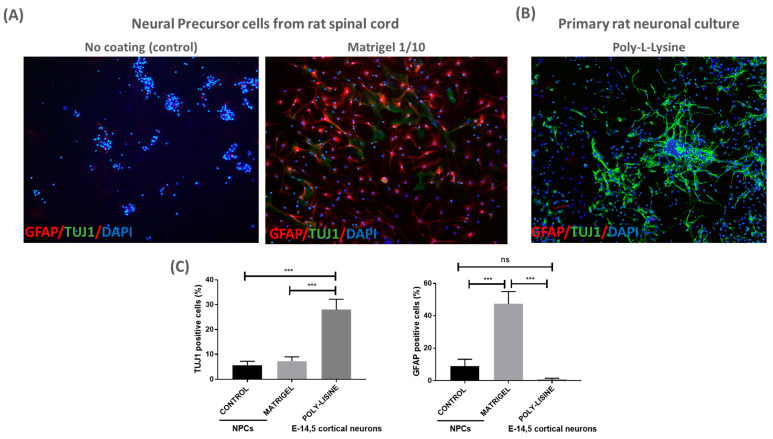
PLA/PCL patches are biocompatible with NPCs culture: (**A**) representative images of NPC-adherent cultures using no coating (control) or Matrigel (1/10) for the PLA/PCL patches. NPC cultures were labeled with GFAP, TUJ1 and DAPI 24 h after cell proliferation over the patch (10× magnification); (**B**) representative image of rat primary neuronal culture coated with poly-L-lysine. Cortical neurons were labeled with GFAP, TUJ1 and DAPI after 3 days in vitro over the patch (10× magnification); and (**C**) graphs show quantification of TUJ1+ and GFAP+ cells detected in NPCs and cortical neurons cultured over the patches. One-way ANOVA test, mean and SEM are shown; ns: non-significant difference, *** *p* ≤ 0.001. n = 3.

**Figure 3 ijms-25-11626-f003:**
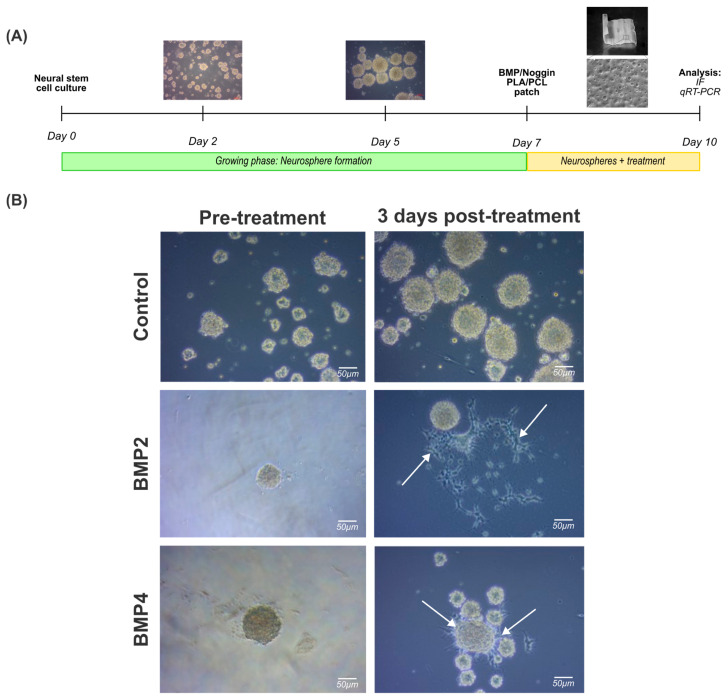
Treatment of 3D neural progenitor cell cultures with BMP2 and BMP4: (**A**) experimental workflow followed to assess the effect of BMP2 or BMP4 on NPC neurospheres; and (**B**) brightfield images documenting the effects of BMP2 or BMP4 on natural spherical development after 3 days post treatment (10 total days in vitro). White arrows indicate deviation from the spherical shape for cells cultured in BMP2 or BMP4 after 3 days post treatment. Scale bar: 50 µm.

**Figure 4 ijms-25-11626-f004:**
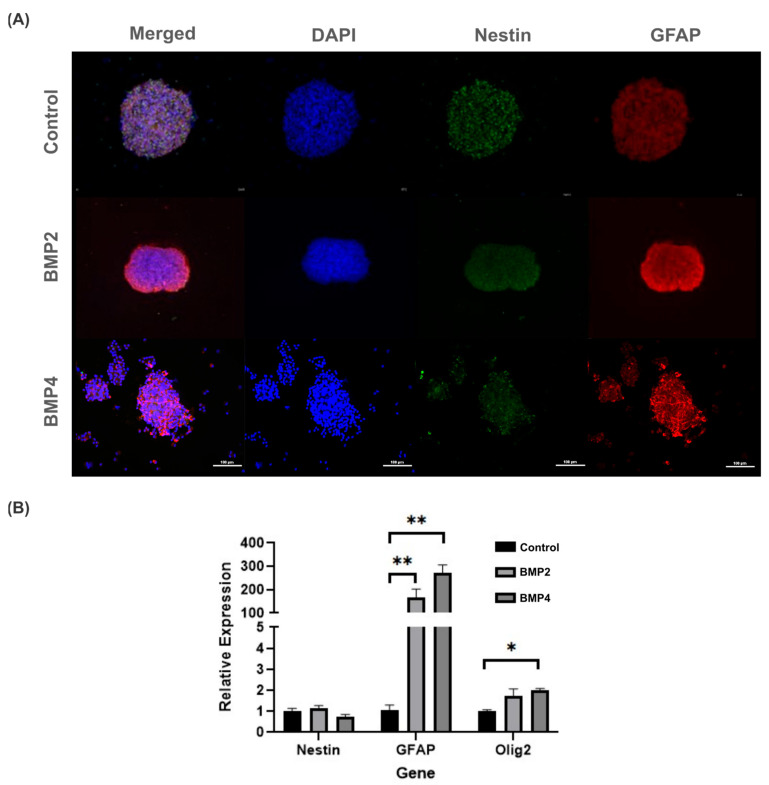
Evaluation of BMP2 and BMP4 effect on NPCs in 3D Neurosphere cultures after 3 days post treatment (10 total days in vitro): (**A**) fluorescence images of NPCs show an increase in GFAP intensity in the presence of BMP2 (middle row) and BMP4 (bottom row), compared to the control (top row). Cell viability was not compromised with BMP treatment. Scale bar: 100 µm; and (**B**) relative gene expression changes in NPCs exposed to BMP2 and BMP4 in 3D NPC cultures. *GFAP* and *Olig2* gene expressions statistically increased after BMP2 and BMP4 exposure, compared to control. Statistical test: one-way analysis of variances (ANOVA) using Tukey’s post hoc test, * *p* ≤ 0.05, ** *p* ≤ 0.01. n = 4 independent cultures.

**Figure 5 ijms-25-11626-f005:**
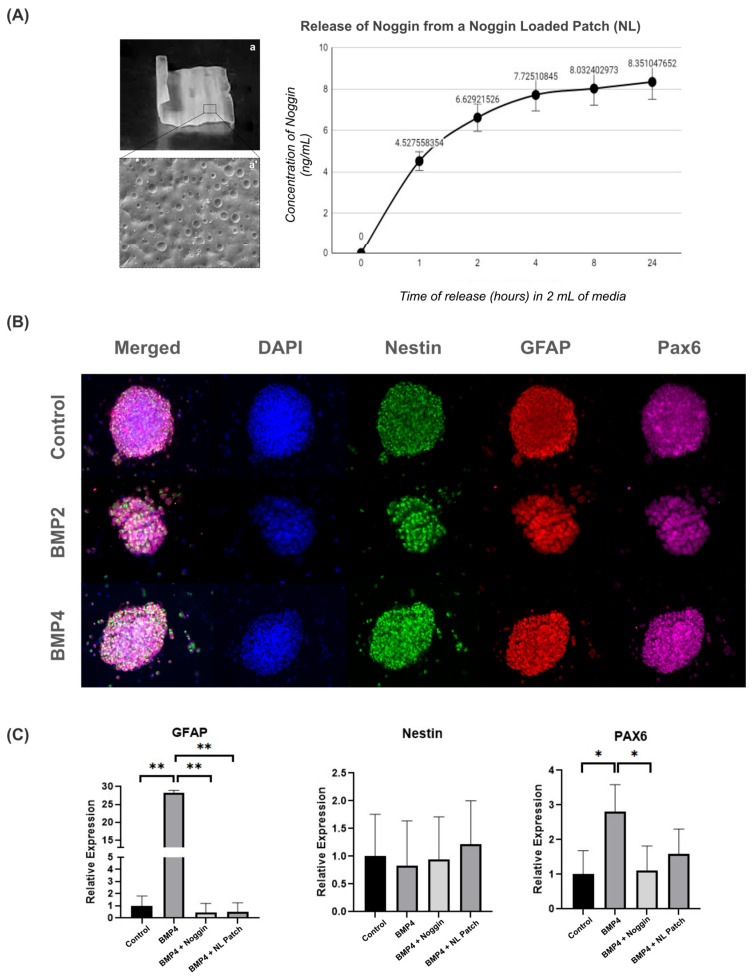
Noggin-loaded patches inhibit the astrogenic differentiation of NPCs: (**A**) image and brief characterization of a 2 × 2 cm^2^ patch (**a**) with an SEM image (5000×) of the surface characteristics (**a’**) and the Noggin release profile following the creation of an NL patch; (**B**) representative images from 3D neurosphere cultures cultured on Noggin-loaded patches, immunostained with progenitor markers (Nestin, GFAP, PAX6) (10× magnification); and (**C**) gene expression changes of progenitor markers from control, BMP4, BMP4 + Noggin and BMP4 + Noggin-loaded (NL) patches. Statistical test: one-way ANOVA, * *p* ≤ 0.05, ** *p* ≤ 0. 01. n = 4 separate wells from 2 independent cultures.

## Data Availability

Data supporting the findings of this study are available within the article.

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
