# Peer review of "Noggin-Loaded PLA/PCL Patch Inhibits BMP-Initiated Reactive Astrogliosis"

_ijms, 2024, doi:10.3390/ijms252111626_

Round 1

Reviewer 1 Report

Comments and Suggestions for Authors

While this study possesses some merit in the presented results, a lot of questions are left unanswered. Most importantly, no material characterisation is conducted on the PLA/PCL material. The authors did not present any mechanical data, no swelling data, no degradation data. All of these factors can heavily influence the response of the cells when in contact with this material. Further, if this material is to be implanted into the patient, how long will it take to be degraded by the body? The addition of the noggin presents an interesting idea, but again no analysis of the diffusion rate of the noggin from the PLA/PCL material is conducted, therefore it is hard to distinguish how much effect the noggin actually has on the cell responses. Including data like this will present a much more comprehensive picture to the reader, not just the in-vitro analysis.

Author Response

We thank the reviewers for their helpful comments and criticisms which made us see the limitations and weaknesses of our study. We followed all the reviewer’s recommendations, and we have carefully addressed each in this revision. Text edits are tracked in the revised manuscript using tracked changes. Please find below a point-by-point response to each reviewer’s comments in which our replies will be in red.  

Reviewer #1

While this study possesses some merit in the presented results, a lot of questions are left unanswered.

Most importantly, no material characterization is conducted on the PLA/PCL material. The authors did not present any mechanical data, no swelling data, no degradation data. All these factors can heavily influence the response of the cells when in contact with this material.

Response: Thank you for your valuable comment. We realized that the explanation in the background section regarding the PLA/PCL patch mechanical properties was insufficient, as these properties have already been published. To address this, we have added a new paragraph that provides more detailed information on the transition temperature, water tightness, and degradation studies conducted in our lab. Please refer to Page 3, lines 98-105 and lines 109-120.

Further, if this material is to be implanted into the patient, how long will it take to be degraded by the body?

Response: Thank you for highlighting this important point regarding our patch, especially as it will be implanted in the body. In response, we have added more details to the background section, referencing our previous in vitro studies that analyzed the degradation of the patch in PBS and amniotic fluid over a period of 16 weeks. A new paragraph has been included, providing additional information about these studies conducted in our lab and previously published. Please see Page 3, lines 98-105 and lines 109-120.

The addition of the noggin presents an interesting idea, but again no analysis of the diffusion rate of the noggin from the PLA/PCL material is conducted, therefore it is hard to distinguish how much effect the noggin actually has on the cell responses. Including data like this will present a much more comprehensive picture to the reader, not just the in-vitro analysis.

Response: We appreciate the reviewer’s comments and have added the requested information in the Methods section (Page 12, lines 455-459). Additionally, the Results section titled 'Noggin-loaded PLA/PCL patches restrict astrogliosis in NPC cultures' has been edited to include the release profile of Noggin into the culture media. Figure 5 has also been revised and is now complete to address this concern (Page 7, 230-237)

Reviewer 2 Report

Comments and Suggestions for Authors

First of all, I congratulate and thank you for getting excellent results and submission.

This study examines whether it can be used as a drug delivery method by controlling the neuronal differentiation process in MMC using a smart patch and BMP. The overall research results are observed to be appropriate. However, some supplements are needed.

1. There is no photo or experimental image of the overall patch, so it will be difficult to understand the contents. Please attach the actual picture of the patch. And add the description of previous results shortly. 

2. As for the previous research results, the PLA/PCL patch is a type according to the degree of expansion, and this is a different process from the drug delivery system, so it has to re-evaluate the cytotoxicity about solvent in PLA/PCL mixture. A study of the effect of BMP should be added. If there is a result of this, request addition, and if there is no, describe why it was omitted.

3. Describe the reason why you chose 3 days for all the results. If you have previous data, please add it. 

4. The experiment was conducted using both BMP2 and BMP4, but the interpretation of the results is unclear as to which one is better. Please state this clearly.

Author Response

We thank the reviewers for their helpful comments and criticisms which made us see the limitations and weaknesses of our study. We followed all the reviewer’s recommendations, and we have carefully addressed each in this revision. Text edits are tracked in the revised manuscript using tracked changes. Please find below a point-by-point response to each reviewer’s comments in which our replies will be in red.

Reviewer 2:

First of all, I congratulate and thank you for getting excellent results and submission.

Response: We thank the reviewer for the comment.

This study examines whether it can be used as a drug delivery method by controlling the neuronal differentiation process in MMC using a smart patch and BMP. The overall research results are observed to be appropriate. However, some supplements are needed.

  1. There is no photo or experimental image of the overall patch, so it will be difficult to understand the contents. Please attach the actual picture of the patch. And add the description of previous results shortly. 

 Response: We apologize for any confusion and appreciate the reviewer’s attention to this matter. Thank you for identifying the issue. New images have been added to the updated Figure 3A, illustrating the workflow, and to Figure 5A, which now includes a picture of the patch unfolding and an SEM image at 5000x magnification. Additionally, a new section has been added to the Introduction, providing details on previously published data regarding the patch characteristics (Please refer to Page 3, lines 98-105 and lines 109-120.).

  1. As for the previous research results, the PLA/PCL patch is a type according to the degree of expansion, and this is a different process from the drug delivery system, so it has to re-evaluate the cytotoxicity about solvent in PLA/PCL mixture. A study of the effect of BMP should be added. If there is a result of this, request addition, and if there is no, describe why it was omitted.

 Response: Thank you for your insightful comment. We have added further clarification regarding the release profile of the patch and the potential cytotoxicity of the mixture. The release profile of Noggin from the NL patch was evaluated in 2 ml of growth media under conditions like those described for the NPC groups (Figure 5A). The majority of Noggin was released within the first 2 hours, reaching an average concentration of 6.63 ± 0.58 ng/ml, with the release rate continuing to decrease over the 24-hour timepoint (Page 7, 230-237). Additionally, previous studies have shown that the PLA/PCL patch exhibits no signs of cytotoxicity or inflammatory effects, both in vitro and in vivo. Furthermore, as the solvent used in preparing the patch is fully evaporated prior to any modifications and use, we have no reason to suspect that the patch adversely affects Noggin during the loading phase (Page 11, lines 359-363).

  1. Describe the reason why you chose 3 days for all the results. If you have previous data, please add it. 

 Response: Thank you for your comments. We believe there was some confusion regarding the timing of the experiments, so we have added more detailed information, including images in the workflow, as shown in the new Figure 3.

As mentioned in the Materials and Methods section, neural stem cells (NSCs) were cultured under standard growth conditions for up to 7 days to allow the neurospheres to reach an optimal size and shape. The growth medium used included NeuroCultTM Proliferation Medium supplemented with NeuroCultTM Proliferation Supplement, 100 U/mL penicillin, 100 µg/mL streptomycin, 0.7 U/mL heparin, 20 ng/mL EGF and 20 ng/mL bFGF. Once the neurospheres reached the appropriate size, we performed BMP2 or BMP4 treatment or noggin-loaded patches evaluation. 

As the reviewer noted here, the data was collected after 3 days post-treatment to assess whether noggin-loaded PLA/PCL smart patches could modulate astrocytic differentiation in neurospheres. This timepoint was selected to avoid masking the true effect of treatment, as neurospheres tend to differentiate into astrocytes, especially in media containing FBS, which is commonly used for differentiation assays in NSC cultures (Lu & Hu, 2009). As observed in Figure 2, neurospheres begin to differentiate shortly after attachment to a coated surface. However, we did not observe NSC differentiation when attached to the patch for the first 3 days (Figure 1). This result led us to consider the 3-day time point as a key window to evaluate the effect of noggin-loaded patches on neurospheres differentiation. 

In addition, the diffusion rate of noggin is mostly during the first few hours after exposure, which made us focus on this short-term response (Figure 5A). Nevertheless, future experiments will be necessary to determine whether noggin-loaded smart patches can modulate NSC differentiation over longer time frames. 

Additionally, we have revised the manuscript and updated the corresponding information in both the text and figure legends to provide greater clarity.

  1. The experiment was conducted using both BMP2 and BMP4, but the interpretation of the results is unclear as to which one is better. Please state this clearly.

Response: We appreciate the reviewer for highlighting this important detail. In response, we have added a paragraph addressing this point in the Discussion section (Page 10, lines 314-346) and included additional references from the literature to support the discussion. Thank you again for your valuable suggestion.

Round 2

Reviewer 1 Report

Comments and Suggestions for Authors

the authors have answered my initial queries with sufficient detail.

Reviewer 2 Report

Comments and Suggestions for Authors

This revised manuscript was well-corrected. I easily understand your point of view and previous results well.